# Molecular Characterization of a Transcriptional Regulator GntR for Gluconate Metabolism in Industrial 2-Ketogluconate Producer *Pseudomonas plecoglossicida* JUIM01

**DOI:** 10.3390/microorganisms13061395

**Published:** 2025-06-15

**Authors:** Mengxin Qu, Lulu Li, Xinyi Zan, Fengjie Cui, Lei Sun, Wenjing Sun

**Affiliations:** 1School of Food and Biological Engineering, Jiangsu University, Zhenjiang 212013, China; 18737741872@163.com (M.Q.); 15380664161@163.com (L.L.); zxy19880920@163.com (X.Z.); fengjiecui@163.com (F.C.); 2Jiangxi Provincial Engineering and Technology Center for Food Additives Bio-Production, Shangrao 334221, China

**Keywords:** 2-ketogluconate (2KGA), Pseudomonas plecoglossicida, LacI family transcriptional regulator GntR, specific binding sites, gluconate metabolism

## Abstract

The GntR is a transcriptional regulator generally known as a gluconate-operon repressor to specifically regulate the transportation and phosphorylation of gluconate. In the present study we report the cloning of the GntR-encoding gene of the industrial 2-ketogluconate (2KGA)-producer *Pseudomonas plecoglossicida* JUIM01, which is involved in the regulation of gluconate metabolism, along with the identification of some of its target genes and its operator sequence. GntR is a 36.36-kDa cytoplasmic and hydrophobic DNA-binding transcriptional regulator belonging to the LacI family. The knockout of *gntR* resulted in the significant upregulation of the transcription of the gluconate kinase gene *gntK* and, to a lesser extent, the permease gene *gntP*, as well as downregulation of genes involved in glucose uptake (*oprB-1*, *gltB*, *gltF*, *gltG*, and *gltK*) and those involved in 2-ketogluconate (2KGA) transport (*kguT*) and catabolism (*kguE*, *kguK*, and *kguD*). These results indicated that GntR positively regulated glucose and 2KGA transport and catabolism, while negatively affecting GntP-mediated gluconate uptake and gluconate phosphorylation by GntK. Electrophoretic mobility shift assay (EMSA) and DNase I footprinting analyses confirmed that GntR interacted with operator sequences in the divergent promoter regions of *gntK* and *gntP*, as well as in the *gntR* promoter region. A putative operator sequence (consensus 5′-AG-N_2_-AGCGCT-N-TCT-3′) was identified. These data suggest that GntR positively regulates genes involved in glucose uptake/transport and 2KGA transport/catabolism, while repressing its own expression as well as that of genes involved in gluconate transport/catabolism. These findings not only elucidate the regulation of GntR and its target genes in *P. plecoglossicida*, but also provide valuable insights for optimizing industrial 2KGA production.

## 1. Introduction

The Embden–Meyerhof–Parnas (EMP) and Entner–Doudoroff (ED) pathways are two abundant dissimilative carbohydrate pathways that form the basis for heterotrophic metabolic chassis strains [1]. Generally, the EMP pathway is regarded as the textbook example of a carbohydrate pathway due to its central role in *Escherichia coli*, *Saccharomyces cerevisiae*, and *Bacillus subtilis* [2]. However, *Pseudomonas* species lack a complete EMP pathway and metabolize glucose primarily via the alternative ED pathway (Figure 1) [3,4,5]. Moreover, *Pseudomonas* strains possess peripheral metabolic pathways that play a critical role in glucose metabolism [6,7]. In these pathways, when glucose enters the periplasmic space, more than 80% of glucose taken up by the cells is successively oxidized to gluconate and 2-ketogluconate (2KGA) by glucose dehydrogenase (Gcd) and gluconate dehydrogenase (Gad), respectively [7,8,9,10]. Glucose, gluconate, and 2KGA in the periplasmic space can enter the cytosol and be metabolized into the key intermediate 6-phosphogluconate (6PG) through three parallel phosphorylation pathways. The 6PG enters the ED pathway, producing 3-phosphoglyceraldehyde (glyceraldehyde-3P) and pyruvate, which are further metabolized in the Krebs cycle to generate energy and precursors for cell growth [3,11]. 2KGA, a product of the peripheral metabolic pathways, is commonly used as an intermediate for synthesizing the GRAS (generally recognized as safe) antioxidant D-isoascorbate (E315-free acid and E316-sodium salt), which helps maintain food color/flavor and prevents the formation of carcinogenic ammonium nitrite during food processing [12,13,14,15]. Based on current knowledge, a diverse set of transcriptional regulators are known to play key roles in regulating the expression of genes encoding catabolic enzymes for glucose metabolism in *Pseudomonas* strains [16], including PtxS [17,18,19,20], PtxR [18,19], HexR [21], GntR/GnuR [22,23], and GtrS-GltR [24,25].

GntR is a transcriptional regulator first identified in *Bacillus subtilis* as a gluconate-operon repressor that specifically regulates the transport and phosphorylation of gluconate [26]. GntR belongs to the GntR-family transcription regulators, a large group of proteins in diverse bacteria that share high similarity in their N-terminal HTH (helix-turn-helix) DNA-binding domains but differ in their C-terminal effector-binding and oligomerization (E-O) domains, enabling them to regulate various biological processes. Based on the heterogeneity of C-terminal effector binding domains and their relationship with corresponding DNA binding domains and *cis*-acting elements (palindromic operator sequences), the GntR family is divided into four main subfamilies (FadR, HutC, MocR, and YtrA) and two minor subfamilies (AraR and PlmA) [27,28,29]. For example, a protein encoded by *sco1678* in *Streptomyces coelicolor* M145 was identified as a GntR-like regulator. It belongs to the FadR subfamily and is involved in the regulation of gluconate metabolism and has a positive impact on actinorhodin biosynthesis and a negative one on coelimycin biosynthesis [30]. In *Streptococcus mutans*, the GntR-family transcriptional regulator StsR (Sugar Transporter Systems Regulator) controls the expression of sugar transporter genes, biofilm formation, and exopolysaccharides synthesis [31]. *B. subtilis* AraR, a repressor of genes involved in L-arabinose metabolism, binds to eight different operators in five different promoter regions with different affinities through its N-terminal DNA-binding domain [32]. *Corynebacterium glutamicum* possesses two functionally equivalent transcriptional regulators, GntR1 and GntR2, which repress genes involved in gluconate metabolism (e.g., *gntK* [gluconate kinase], *gntP* [gluconate permease], and *gnd* [6-phosphogluconate dehydrogenase]), while activating *ptsG*, encoding the EII^Glc^ permease of the glucose phosphotransferase system (PTS), thereby in an opposite way regulating gluconate uptake and catabolism and glucose uptake [33]. In *Vibrio cholerae*, GntR negatively regulates the gluconate utilization system by controlling *gntU* (gluconate transporter) and *gntK* (gluconate kinase) expression. The ability of *V. cholerae* to utilize gluconate directly affects its colonization potential and cholera toxin secretion levels [34]. In *E. coli*, the *gntK*, *gntU*, and *gntT* constitute the GntI gluconate utilization system that is negatively regulated by the constitutively expressed GntR [35,36].

Studies on the transcriptional regulator GntR in *Pseudomonas* species have primarily focused on *P. aeruginosa* and *P*. *putida*. For instance, GntR in *P. aeruginosa* has been shown to repress its own expression as well as that of the GntP gluconate permease. GntR is released from its target site in the presence of gluconate and 6-phosphogluconate, which bind with similar apparent affinities to the GntR-DNA complex to regulate the glucose metabolism [22]. However, few reports exist regarding GntR in other *Pseudomonas* species, particularly industrial strains, as evidenced by literature reviews and database searches (NCBI, and Pseudomonas Genome Database: https://www.pseudomonas.com/, accessed on 9 April 2025). In consequence, the present study aimed to elucidate the role of GntR in glucose/gluconate metabolism by the following: (1) identifying and heterologously expressing *gntR* from the industrial 2KGA-producer *P. plecoglossicida* JUIM01 [37]; (2) characterizing the molecular features of the purified protein using MALDI-TOF-MS and circular dichroism (CD) spectroscopy; and (3) identifying potential target genes involved in glucose/gluconate metabolism through *gntR* knockout, electrophoretic mobility shift assays (EMSA), and DNase I footprinting analyses. The findings of this study provide preliminary insights into the role of GntR in glucose/gluconate metabolism in the industrial 2KGA producer *P. plecoglossicida* JUIM01, establishing a theoretical foundation for optimizing 2KGA production.

## 2. Materials and Methods

### 2.1. Bacterial Strains, Plasmids, and Media

The bacterial strains and plasmids used in this study are listed in Table 1. *P. plecoglossicida* JUIM01, screened and maintained in our laboratory, is an industrial 2KGA production strain in China. Unless otherwise specified, *E. coli* strains were cultivated at 37 °C in LB medium (10.0 g/L of tryptone, 5.0 g/L of yeast extract, and 10.0 g/L of NaCl) or on LB agar plates. When required, antibiotics were supplemented at the following concentrations: 50 μg/mL ampicillin or 25 μg/mL kanamycin sulfate. For *P. plecoglossicida* cultivation, cells were initially activated on the medium containing 10 g/L peptone, 5 g/L beef extract, 5 g/L NaCl and 20 g/L agar (pH 7.0). Activated cells were then inoculated into 50 mL of seed medium (20 g/L glucose, 5 g/L corn syrup powder, 2 g/L urea, 2 g/L KH_2_PO_4_, 0.5 g/L MgSO_4_·7H_2_O and 5 g/L CaCO_3_, pH 7.0) in 500-mL Erlenmeyer flask and cultured at 33 °C with shaking at 265 rpm.

### 2.2. Bioinformatics Analyses of the gntR Gene and GntR Protein in P. plecoglossicida JUIM01

Genome DNA of *P. plecoglossicida* JUIM01 was extracted using a Bacterial Genomic DNA Kit (Beyotime, Shanghai, China). The *gntR* gene was amplified by PCR using primers *gntR*-F and *gntR*-R (Appendix A) with genomic DNA as template. The PCR product was sequenced by Sangon Biotech Co. (Shanghai, China). ORF Finder (https://www.ncbi.nlm.nih.gov/orffinder/, accessed on 9 April 2025) was used to predict the open reading frame of *gntR*. NCBI tools (https://blast.ncbi.nlm.nih.gov/Blast.cgi, accessed on 9 April 2025) were used to analyze the identity of *gntR* and encoded protein sequence. The conserved domain of GntR was predicted using the Conserved Domain Search Service tool (CD Search) (https://www.ncbi.nlm.nih.gov/Structure/cdd/wrpsb.cgi, accessed on 9 April 2025). The physio-chemical properties of molecular weight, PI, amino acid compositions, and hydropathy plot of GntR were predicted using ProtParam tool (https://web.expasy.org/protparam/, accessed on 9 April 2025). The TMHMM Server 2.0 (http://www.cbs.dtu.dk/services/TMHMM-2.0, accessed on 9 April 2025), SignalP 5.0 (https://services.healthtech.dtu.dk/services/SignalP-5.0/, accessed on 9 April 2025), and SOPMA (https://npsa-prabi.ibcp.fr/cgi-bin/npsa_automat.pl?page=npsa_sopma.html, accessed on 9 April 2025) were used to analyze the structural features of GntR including the transmembrane structure, signal peptides, and secondary structure, respectively. AlphaFold 3 model (https://alphafoldserver.com/, accessed on 9 April 2025) and LigPlot^+^ v.2.2 (EMBL-EBI, UK) were used to predict the tertiary structure of the GntR protein (as a monomer and a dimer) and its binding to the specific DNA sequence of 5′-AG-N_2_-AGCGCT-N-TCT-3′. BPROM (http://www.softberry.com/berry.phtml?topic=bprom&group=programs&subgroup=gfindb, accessed on 9 April 2025) and BDGP (http://www.fruitfly.org/seq_tools/promoter.html, accessed on 9 April 2025) were used to predict promoters.

### 2.3. Construction of the Recombinant Strain to Express GntR

The full-length *gntR* was PCR-amplified from *P. plecoglossicida* JUIM01 genomic DNA using primers *gntR*-F and *gntR*-R (Appendix A) and subsequently cloned into the pMD20-T vector via TA cloning, yielding the construct pMD20-T-*gntR*. Following digestion with *Nde* I and *Xho* I restriction enzymes, the *gntR* fragment was ligated into similarly digested pET-28a(+) to generate the expression plasmid pET-28a(+)-*gntR*. The pET-28a(+)-*gntR* was then transformed into *E. coli* BL21(DE3)-competent cells to create the heterologous expression strain *E. coli* BL21(DE3)/pET-28a(+)-*gntR*. Positive transformants were selected on LB agar plates containing 25 µg/mL kanamycin and subsequently cultivated in LB medium for verification. Plasmid extraction, *Nde* I/*Xho* I restriction analysis, and sequencing confirmed the correct constructs (Appendix A).

### 2.4. Expression and Purification of the Recombinant GntR Protein (PpGntR)

The recombinant *E. coli* BL21(DE3)/pET-28a(+)-*gntR* strain was cultured overnight and then inoculated (1% inoculum) into LB medium containing 25 μg/mL kanamycin. Cultures were grown at 37 °C with 200 rpm shaking until reaching an optical density at 600 nm (OD_600nm_) of 0.5–1.0. Protein expression was induced by adding isopropyl-β-D-thiogalactoside (IPTG) to final concentrations of 0.2–1.0 mM (in 0.2 mM increments), followed by overnight incubation at 20 °C with 200 rpm shaking. Recombinant GntR (PpGntR) expression was verified by SDS-PAGE and Western blot analysis using an anti-6×His rabbit polyclonal antibody (primary antibody) and HRP-conjugated goat anti-rabbit IgG (secondary antibody).

Cells were harvested by centrifugation (5000× *g*, 10 min, 4 °C) and washed twice with 100 mM phosphate-buffered saline (PBS, pH 7.4). The cell pellets were resuspended in 40 mL of buffer A (20 mM Tris-HCl, 500 mM NaCl, 1 mM DTT, 20 mM imidazole, pH 7.9) and lysed by ultrasonication (30 min). The lysates were centrifuged (12,000× *g*, 30 min, 4 °C) to separate soluble and insoluble fractions. The supernatant was loaded onto a 5 mL HisTrap™ HP column and eluted with a linear imidazole gradient (100–500 mM) prepared by mixing buffer A and buffer B (20 mM Tris-HCl, 500 mM NaCl, 1 mM DTT, 500 mM imidazole, pH 7.9). Optimal elution conditions were determined by SDS-PAGE analysis. Selected fractions were dialyzed overnight at 4 °C against 50 mM HEPES buffer (300 mM NaCl, 1 mM DTT, 10% glycerol, pH 7.9) and concentrated using a 10-kDa molecular weight cutoff centrifugal filter (MilliporeSigma, Burlington, MA, USA) at 5000× *g* and 4 °C. Purified PpGntR was stored at −80 °C for subsequent experiments.

### 2.5. Matrix-Assisted Laser Desorption/Ionization Time of Flight Mass Spectrometry (MALDI-TOF-MS)

The purified protein was identified using a MALDI-TOF/TOF mass spectrometer (4800 Plus MALDI TOF/TOF Analyzer, AB Sciex, Framingham, MA, USA) equipped with a 200-Hz frequency Nd:YAG laser (355 nm wavelength) with a mass scanning range of 1000–80,000 Da. Spectra were acquired using pulsed ion extraction with a 1300 ns delay time. Measurements were performed in linear positive ion mode with an acceleration voltage of 19.4 kV (Grid 1) and a lens voltage of 8 kV. External calibration was performed using 0.5 mg/mL myoglobin as standard. For MALDI-TOF-MS analysis, 2 µL of PpGntR sample was mixed with an equal volume of matrix solution [12 mg/mL α-cyano-4-hydroxycinnamic acid (CHCA; Sigma-Aldrich, St. Louis, MO, USA) in 100% acetonitrile containing 0.3% trifluoroacetic acid]. The mixture was spotted onto a stainless steel MALDI target plate (123 × 81 mm Opti-TOF 384-well insert; AB Sciex). Mass spectra were processed using flexAnalysis software v 3.4 (Bruker Daltonics Inc., Billerica, MA, USA).

### 2.6. Circular Dichroism (CD) Far-Ultraviolet Scanning Analysis

The secondary structure of PpGntR was analyzed by far-UV circular dichroism (CD) spectroscopy using a Chirascan™ Plus CD spectrometer (Applied Photophysics Ltd., Leatherhead, UK). Measurements were performed in 0.1 cm pathlength quartz cuvettes at room temperature, scanning in the range of 190–260 nm with the following parameters: 0.1 nm data interval, 100 nm/min scan speed, 1 nm bandwidth, 1 s response time, and 15 accumulations. For calibration, far-UV CD spectra of camphorsulfonic acid (CSA) standard solution (180–340 nm) were acquired. Sample spectra were collected under identical conditions and processed using Pro-Data Viewer software v 4.2 (Applied Photophysics Ltd., Leatherhead, UK). Secondary structure content was calculated from the CD spectra using CDNN analysis software v 2.1 (Applied Photophysics Ltd., Leatherhead, UK).

### 2.7. Construction of the gntR-Deleted and gntR-Complemented Recombinants Derived from P. plecoglossicida JUIM01

The *gntR*-deleted mutant of *P. plecoglossicida* JUIM01 (named as JUIM01Δ*gntR*) was constructed using plasmid pK18mobsacB via a two-step homologous recombination method [38]. Two homologous arms flanking the *gntR* gene were amplified from JUIM01 genomic DNA using primer pairs Δ*gntR*-K1/Δ*gntR*-K2 and Δ*gntR*-K3/Δ*gntR*-K4. These fragments were then fused by overlap extension PCR using primers Δ*gntR*-K1/Δ*gntR*-K4, digested with *Eco*R I and *Hin*d III, and ligated into similarly digested pK18*mobsacB* to generate the recombinant suicide plasmid pK18*mobsacB*-Δ*gntR* (Appendix A). The plasmid was electroporated into freshly prepared JUIM01 competent cells. Mutants were selected through two rounds of homologous recombination: first on LB plates containing 25 μg/mL kanamycin, followed by counter-selection on LB agar plates with 10% sucrose. Successful *gntR* deletion in *P. plecoglossicida* JUIM01Δ*gntR* was confirmed by colony PCR (Appendix A) and sequencing. The corresponding *gntR-complemented* strain (named as JUIM01Δ*gntR-gntR*) was constructed by transforming the recombinant plasmid pBB-*gntR* (the expression vector pBBR1MCS-2 linked with *gntR*) into JUIM01Δ*gntR*.

### 2.8. Quantitative Real-Time PCR (RT-qPCR)

Total RNAs were extracted from *P. plecoglossicida* JUIM01 and *P. plecoglossicida* JUIM01Δ*gntR* using TRIzol reagent (Vazyme Biotech, Nanjing, China). First-strand cDNA was synthesized from the total RNAs using HiScript II Q RT SuperMix for qPCR kits (+gDNA wiper) (Vazyme Biotech, Nanjing, China). RT-qPCR analysis was performed using a 7300 real-time PCR system (Applied Biosystems, Carlsbad, CA, USA) with ChamQ SYBR Color qPCR Master Mix (2×) (Vazyme Biotech, Nanjing, China) following the manufacturer’s instruction. Each 10 μL reaction mixture contained 5.0 μL 2× ChamQ SYBR Color qPCR Master Mix, 0.4 μL the primers used for RT-qPCR (5 µM) (Appendix A), 0.2 μL 50× ROX Reference Dye 1, 1.0 μL cDNA, and 3.0 μL nuclease-free water. The thermal cycling conditions consisted of initial denaturation at 95 °C for 3 min, and 40 cycles of denaturation at 95 °C for 5 s, annealing at 55 °C for 30 s, and extension at 72 °C for 40 s. Gene expression levels were normalized to 16S rRNA and calculated using the 2^−△△Ct^ method with three biological replicates.

### 2.9. Electrophoretic Mobility Shift Assay (EMSA)

The recombinant plasmids pUC57-P*_gntR_,* pUC57-P*_gntK_*, and pUC57-P*_gntP_* containing the promoter regions of *gntR*, *gntK*, and *gntP*, respectively, were constructed (Appendix A). 5′-Carboxyfluorescein (FAM)-labeled DNA probes for P*_gntR_* (278 pb), P*_gntK_* (280 pb), and P*_gntP_* (288 pb) were PCR-amplified using the respective pUC57 constructs as templates with M13-F (5′-GTTGTAAAACGACGGCCAG-3′, FAM-labeled) and M13-R (5′-CAGGAAACAGCTATGAC-3′) as primers. The 5′-FAM-labeled PCR products were verified by 2% agarose gel electrophoresis and purified using a DiaSpin PCR Product Purification Kit (Sangon, Shanghai, China).

For the EMSA binding assay, 20 µL reaction systems contained 50 ng FAM-labeled promoter probe (P*_gntR_*, P*_gntK_* or P*_gntP_*), 0, 5, or 10 µg purified PpGntR, 2 µL 10× binding buffer, 2 µg salmon sperm DNA (as a nonspecific cold DNA probe), and nuclease-free water. Competitive binding reactions included 5 μg PpGntR, 50 ng FAM-labeled probe, and 5 μg of unlabeled competitor DNA (cold probe). After 30 min incubation at 25 °C, reaction products were resolved on native 6% polyacrylamide gels in 0.5× TBE buffer (45 mM Tris, 45 mM boric acid, 1 mM EDTA, pH 8.3). Fluorescent DNA–protein complexes were visualized using a Typhoon Trio system (GE Healthcare Bio-Sciences, Pittsburgh, PA, USA).

### 2.10. DNase I Footprinting Assay

The 5′-FAM-labeled DNA probes (P*_gntR_*, P*_gntK_* or P*_gntP_*) were incubated with PpGntR in 40 µL reaction mixtures containing 150–160 ng DNA probe, 0 or 10 µg PpGntR, 4 µL 10× binding buffer, 4 µg salmon sperm DNA, and nuclease-free water. After 30 min at 25 °C, the samples were digested with DNase I in 100 µL reactions containing 10 µL 10× RQI buffer, 1 µL 1 M CaCl_2_, 0.2 µL DNase I, and nuclease-free water. After 55 s digestion at 37 °C, reactions were terminated by adding 10 µL 0.5 M EDTA and incubating at 65 °C for 10 min. Digestion products were separated by capillary electrophoresis, sequenced using an ABI 3730XL Genetic Analyzer (Foster City, CA, USA). Peak Scanner software v1.0 (Applied Biosystems) was used for fluorescence peaks and base calling.

### 2.11. Statistical Analysis

All experiments included three biological replicates. Data are presented as mean ± standard deviation (*n* = 3) and analyzed by one-way analysis of variance (ANOVA). Statistical significance was determined on the basis of the *p* value (*α* = 0.05 and 0.01).

## 3. Results and Discussion

### 3.1. Cloning and Identification of the gntR Gene of P. plecoglossicida JUIM01

Analysis of the Pseudomonas Genome Database (https://www.pseudomonas.com/, accessed on 9 April 2025) revealed 483 annotated *gntR* genes (1029–1032 bp) across *P. aeruginosa*, *P. syringae*, *P. protegens*, and *P. stutzeri*. However, no *gntR*-like gene originating from *P. plecoglossicida* was present in this database. In this study, a 1020-bp full sequence of *gntR* was cloned from *P. plecoglossicida* JUIM01 using *gntR*-F and *gntR*-R as primers. Sequence analysis showed that the *gntR* of *P. plecoglossicida* JUIM01 shared 86.98%, 87.12%, 87.12%, 87.22%, 87.28%, and 98.14% identity to those of *P. mosselii* (CP024159.1), *Pseudomonas* sp. 2hn (CP081016.1), *Pseudomonas* sp. LH21 (CP144367.1), *Pseudomonas* sp. RtIB026 (AP023348.2), *P. mosselii* (CP104107.1), and *Pseudomonas* sp. *p1* (CP083746.1), respectively. The encoded GntR (339 aa, 36.36 kDa) was characterized as a cytoplasmic hydrophobic protein (Appendix A) with high similarity to LacI family regulators, sharing 98.82%, 98.82%, 99.12%, 99.12%, 99.71%, and 100.00% identity to *Pseudomonas* sp*. p1* (WP_224455985.1), *P. putida* (WP_084858081.1), *P. putida* (EKT4521039.1), *P. putida* (EKT4467508.1), *P. juntendi* (WP_368828869.1), and *P. putida* (WP_324815481.1), respectively. Domain analysis classified *P. plecoglossicida* GntR as a LacI-family transcriptional regulator featuring a conserved N-terminal HTH (helix-turn-helix) DNA-binding domain and a C-terminal gluconate-specific ligand-binding domain. Secondary structure prediction (SOPMA) indicated that *P. plecoglossicida* GntR consisted of 44.25% of α-helix, 16.81% of extended strand, 6.78% of β-turn, and 32.15% of random coil.

### 3.2. Heterogeneous Expression and Purification of the Recombinant GntR Protein (PpGntR)

The recombinant *E. coli* BL21(DE3)/pET-28a(+)-*gntR* strain was constructed for the heterologous expression of *P. plecoglossicida* GntR, with *E. coli* BL21(DE3)/pET-28a(+) serving as a negative control. The recombinant strains were cultured and induced by different concentrations of IPTG. Following induction with varying IPTG concentrations (0.2–1.0 mM), SDS-PAGE (Appendix A) analysis revealed the following: no detectable GntR expression in uninduced *E. coli* BL21(DE3)/pET-28a(+)-*gntR* or the control *E. coli* BL21(DE3)/pET-28a(+). Distinct ~38 kDa bands, corresponded to His-tagged recombinant PpGntR, were observed in IPTG-induced samples (Lanes 3–7), closely matching the theoretical molecular weight (38.67 kDa) and the annotated GntR proteins (37.1 kDa) from *P. aeruginosa* PAO1 and PAK [39,40]. Western blot confirmation was performed for 0.4 mM IPTG-induced PpGntR (Appendix A). Cellular fractionation demonstrated soluble expression, with PpGntR predominantly present in the supernatant. Ni-NTA affinity chromatography yielded high-purity protein, as evidenced by a single band on SDS-PAGE with optimal elution at 200–250 mM imidazole (Appendix A).

### 3.3. Molecular Weight and Secondary Structure of PpGntR

Matrix-assisted laser desorption/ionization (MALDI) is suited for the ionization of high molecular weight compounds, and time of flight mass spectrometry (TOF-MS) allows for the determination of their exact molecular masses. As shown in Appendix A, PpGntR exhibited a strong signal at *m*/*z* of 38,341, corresponding to a molecular weight of approximately 38.34 kDa. The result closely matched the predicted molecular weight of 38.67 kDa from bioinformatics analyses. The secondary structure of purified PpGntR was analyzed by circular dichroism (CD) spectroscopy. The contents of α-helix, β-sheet (anti-parallel and parallel), β-turn, and random coil in different wavelength intervals were calculated in the milli-degrees mode using CDNN software v 2.1 [41,42]. The analysis revealed that the composition of PpGntR included 33.8%, 33.1%, 16.3%, and 13.5% of α-helix, random coil, β-turn, and antiparallel & parallel, respectively, within the wavelength of 190–260 nm (Appendix A). Compared to the predicted secondary structure, a decrease in α-helical content and an increase in β-sheet content were observed.

### 3.4. Effect of gntR Gene Deletion on Expression of Glucose/Gluoconate-Metabolism-Related Genes in P. plecoglossicida JUIM01

Based on previous findings, *gntP* (encoding gluconate permease) and *gntK* (also annotated as *gnuK*, encoding gluconate kinase) are identified as responsible for gluconate transport and phosphorylation, respectively. The *gcd* encoding glucose dehydrogenase is responsible for oxidization of glucose to gluconate. The *gad* operon (composed of *gadA*, *gadB*, and *gadC*) encoding gluconate dehydrogenase is responsible for 2KGA synthesis. The genes *kguE*, *kguK*, *kguT*, and *kguD* encoding a putative epimerase, 2KGA kinase, 2KGA transporter, and 2-keto-6-phosphogluconate reductase are involved in 2KGA transport and catabolism. Meanwhile, *oprB-1* (encoding glucose uptake porin) and the *gltBFGK* operon (encoding ABC glucose transporter subunits) mediate glucose uptake [18,19,20,35,43]. A strain of *P. plecoglossicida* JUIM01 deleted for the transcriptional regulator GntR (JUIM01Δ*gntR*) was constructed in order to assess the impact of GntR on glucose uptake and metabolism. Comparative qRT-PCR analysis evaluated transcript levels of 14 target genes (*gntK*, *gntP*, *ptxS*, *kguE*, *kguK*, *kguT*, *kguD, gltR*, *gtrS*, *gltB*, *gltF*, *gltG*, *gltK*, and *oprB-1*) in wild-type (WT) versus *gntR*-deleted strains after 12 h seed culture using 16S rRNA gene as the housekeeping gene. As shown in Figure 2, the expression levels of *gntK* and *gntP* were significantly upregulated (*p* < 0.0001), 6.73- and 2.23-fold, respectively, in JUIM01Δ*gntR* compared to those in WT. This indicated that GntR negatively regulates the expression of these genes. This result is consistent with previous reports that GntR of *P. aeruginosa* acts as a repressor of gluconate metabolism genes [22,26].

In contrast, expression of *oprB-1*, *gltB, gltF, gltG*, and *gltK* was significantly downregulated (6.52- to 11.78-fold), along with genes of the *kguE/K/T/D* operon (*kgu* operon [43,44]) which showed a 3.12- to 4.53-fold downregulation compared to WT. These results suggested that GntR might positively regulate the expression of these genes. This suggested that glucose uptake/transport and 2KGA transport and catabolism might be negatively affected in JUIM01Δ*gntR* whereas gluconate transport and phosphorylation might be stimulated. The relative expression levels of *gcd* and the *gad* operon (*gadA*, *gadB*, and *gadC*) responsible for the conversion of glucose to gluconate and then to 2KGA were modestly decreased (1.22-, 1.34-, 1.55-, and 1.49-fold, respectively) in JUIM01Δ*gntR*. Interestingly, the expression of *ptxS*, *gtrS*, and *gltR* was also modestly decreased (1.32-, 1.41-, and 1.28-fold, respectively). PtxS is a transcriptional regulator which specifically binds to a 14-bp palindrome (TGAAACCGGTTTCA) and negatively regulates the expression of genes involved in 2KGA metabolism, including the *gad* operon encoding gluconate dehydrogenase (Gad), the *kgu* operon, and its own gene [17,18,19,20]. In *P. plecoglossicida* JUIM01, this palindrome or similar sequence was identified upstream of the *kgu* operon and *ptxS*. However, no such sequence was found in the promoter region of the *gad* operon. *gtrS* together with *gltR* encode a two-component regulatory system GtrS-GltR, which regulates the target genes including *gltB* and is required for glucose transport in *P. aeruginosa* [24,25]. Daddaoua et al. reported that the GltR operator site contains the consensus sequence tgGTTTTTc [24]. However, the potential regulatory site of GltR identified by Xu et al. shows greater diversity [25]. GtrS-GltR has not been reported in *P. plecoglossicida*, but sequences such as CTTTTTC, CTTTCTTGTTGTTATG, and CTTTTTG were found near the start codon of *oprB-1*, *gltB*, and *oprB-2* (the first structural gene of the operon where *gcd* is located). Whether and how these genes are regulated by GtrS-GltR in *P. plecoglossicida* requires further investigation. However, further study indicated that the deletion or complementation of *gltR* had a minimal impact on growth of the strains in the presence of glucose (Appendix A). This might be due to the existence of a complex regulatory network governing *Pseudomonas* carbon metabolism, where multiple transcriptional regulators (such as GntR, PtxS, and GtrS-GltR) coordinately control substrate (glucose, gluconate, and 2KGA) transport, metabolic flux partitioning, and cellular growth homeostasis [7,45].

In *P. plecoglossicida* JUIM01, over 80% of glucose can be converted to gluconate, which then undergoes two metabolic fates: further oxidation to 2KGA, or transport into cells via GntP followed by phosphorylation via GntK. The transcriptional repression of *gntK* and *gntP* by GntR promotes carbon flux toward 2KGA production, while upregulated expression of the *kgu* operon conversely limits 2KGA accumulation. Therefore, a potential metabolic engineering strategy for *P. plecoglossicida* JUIM01 to improve 2KGA accumulation would involve the following: (1) over-expression of GntR to repress the expression of *gntK* and *gntP*, thereby reducing gluconate that enters the cytoplasm for catabolism; and (2) inhibition of the expression of the *kgu* operon through gene knockout, RNA interference, or promoter engineering to render the *kgu* operon under GntR regulation, to prevent 2KGA from entering the cytoplasm for catabolism.

### 3.5. Specific Binding of PpGntR to the Promoter Region of the Potential Target Genes gntK and gntP

Based on the transcriptional changes observed in the *gntR* deletion mutant, it was inferred that *gntK* and *gntP* were potential GntR target genes. As shown in Figure 2, *gntR* is located upstream and divergently from *gntK*, with a 208-bp intergenic promoter region. *gntP* is located 100-bp downstream of *gntK* and is transcribed in the same direction. This arrangement supports the common regulatory paradigm where transcriptional regulators frequently autoregulate their own expression, and regulatory genes are located in close vicinity of their targets genes [44,46,47].

The transcriptional regulators have the ability to recognize and bind to short specific DNA sequences located either upstream or in overlap of promoter regions that control the initiation of transcription and thus regulate gene expression [48]. EMSA validation was performed to characterize PpGntR–DNA interactions using 50 ng of FAM-labeled DNA probe (P*_gntR_*, P*_gntK_* or P*_gntP_*), 0 μg, increasing PpGntR concentrations (0, 5, and 10 μg), and 100× cold probe competition. The results (Figure 3) revealed specific interactions between PpGntR and its target promoters, with control lanes (Lane 1/5/9) showing unshifted migration of FAM-labeled P*_gntR_*, P*_gntK_*, and P*_gntP_* probes in the absence of PpGntR. Dose-dependent band shifts were observed upon addition of increasing PpGntR concentrations (5 μg in Lane 2/6/10; 10 μg in Lane 3/7/11), demonstrating protein–DNA complex formation. Binding specificity was further validated through competitive displacement assays (Lane 4/8/12), where 100-fold molar excess of unlabeled probe competitively bound most PpGntR, thereby restoring the migration of the free labeled probes. Additionally, the system contained labeled probes that were unsaturatedly bound to PpGntR. The migration rate of these protein–DNA complexes is slower than that of the free probes but faster than that of the saturated protein–DNA complexes, causing them to appear in the middle position of the lanes. These findings conclusively established that PpGntR specifically interacts with specific operator sequences located in the intergenic regions located between *gntR* and *gntK* and in the *gntP* promoter regions.

### 3.6. Identification of the Binding Sites of PpGntR to the Promoter Region of the Target Genes

DNase I footprinting analyses identified distinctly protected sequences in the promoter regions of *gntR*, *gnuK,* and *gntP* of *P. plecoglossicida* JUIM01 when 10 μg of PpGntR was present (Figure 4A–C). The protected DNA sequence in P*_gntR_* (5′-TAAGGTAGCGCTGTCTCAGGACAC-3′) was 24-bp long, and those of P*_gnuK_* (5′-GCGAGACAGCGCTATCTTACCC-3′) and P*_gntP_* (5′-CAAAGACAGCGCTGTCTCGACA-3′) were 22-bp long. Bioinformatic alignment identified a conserved binding motif (5′-AG-N_2_-AGCGCT-N-TCT-3′) shared by all three promoters (Figure 4D,E). Structural modeling using AlphaFold 3 and LigPlot^+^ predicted that the 5′-AG-N_2_-AGCGCT-N-TCT-3′ motif could bind to the HTH domain of *P. plecoglossicida* GntR (Figure 5), consistent with the DNA-binding characteristics of LacI-family transcriptional regulators [26,48]. Notably, *P. aeruginosa* PAO1 GntR recognizes a similar sequence (5′-AC-N-AAG-NTAGCGCT-3′) in its *gntR* and *gntP* promoters, where the binding sites overlap with the −10 promoter elements, which consequently enables GntR to inhibit transcription by blocking RNA polymerase binding [22]. The −10 and −35 regions of P*_gntR_*, P*_gntK_*, and P*_gntP_* were predicted using bioinformatics tools (Figure 4E). The predicted −10 regions (RNA polymerase recognition sites) of the P*_gntR_* and P*_gnuK_* in *P. plecoglossicida* JUIM01 overlapped with the above same binding sequence while the predicted −10 region of the P*_gntP_* had no similar overlapped sequence (Figure 4E). In this regard, our speculations are the following: (1) another possible -10 sequence is the consensus palindromic sequence AGCGCT found in P*_gntR_*, P*_gntK_*, and P*_gntP_*, ensuring that the −10 regions overlap with the GntR binding sites; (2) *gntP* and the upstream *gntK* are likely located in the same operon and are co-transcribed, meaning there is no −10 sequence in the 100-bp intergenic region between *gntK* and *gntP*, which is common in the regulatory mechanisms of LacI-family transcriptional regulators. The speculations need to be verified by further studies. These results suggested that GntR proteins in genus *Pseudomonas* might employ similar regulatory mechanisms.

## 4. Conclusions

This study reports the first cloning of a 1020-bp gene *gntR* from *P. plecoglossicida* JUIM01, which encoded a 36.36-kDa cytoplasmic and hydrophobic polypeptide comprising 339 amino acids belonging to the LacI family of DNA-binding transcriptional regulators. The secondary structure of the recombinant PpGntR was characterized, and the tertiary structure of GntR of *P. plecoglossicida* JUIM01 was predicted. Knockout of *gntR* demonstrated that GntR functions as a global regulator, negatively controlling its own expression as well as that of the genes involved in gluconate metabolism (*gntP* and *gntK*), while positively regulating glucose uptake systems (*oprB-1*, *gltB/F/G/K*) and 2KGA catabolic genes (the *kgu* operon). This specific regulatory pattern reflected the global regulatory role of GntR in glucose/gluconate/2KGA transport and metabolism. EMSA and DNase I footprinting analyses results confirmed that the PpGntR bound to its own promoter region and to that of *gntP* and *gntK*. A consensus operator site (5′-AG-N_2_-AGCGCT-N-TCT-3′) shared by all three promoters was identified. These findings not only elucidate the transcriptional control mechanisms governing carbon metabolism in *P. plecoglossicida*, but also provide valuable insights for optimizing industrial 2KGA production.

## Figures and Tables

**Figure 1 microorganisms-13-01395-f001:**
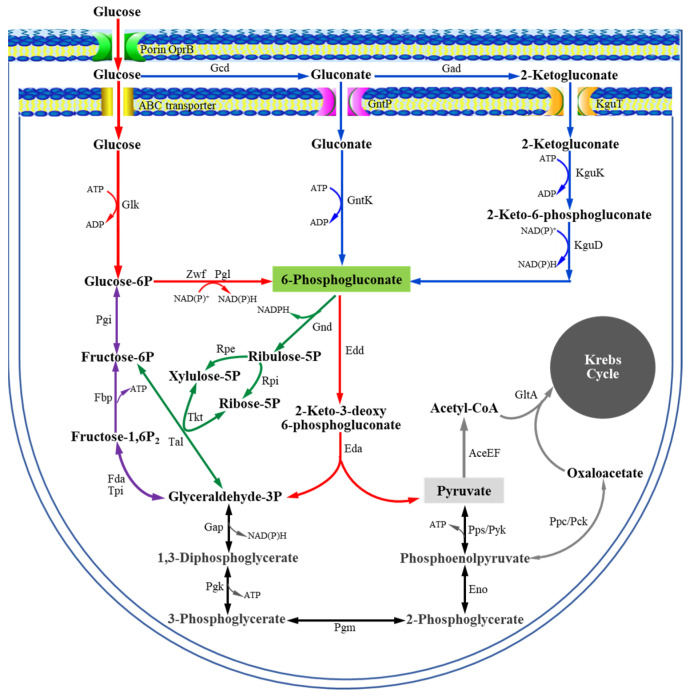
Glucose metabolism in *Pseudomonas* deduced from gene annotations and functional analysis.

**Figure 2 microorganisms-13-01395-f002:**
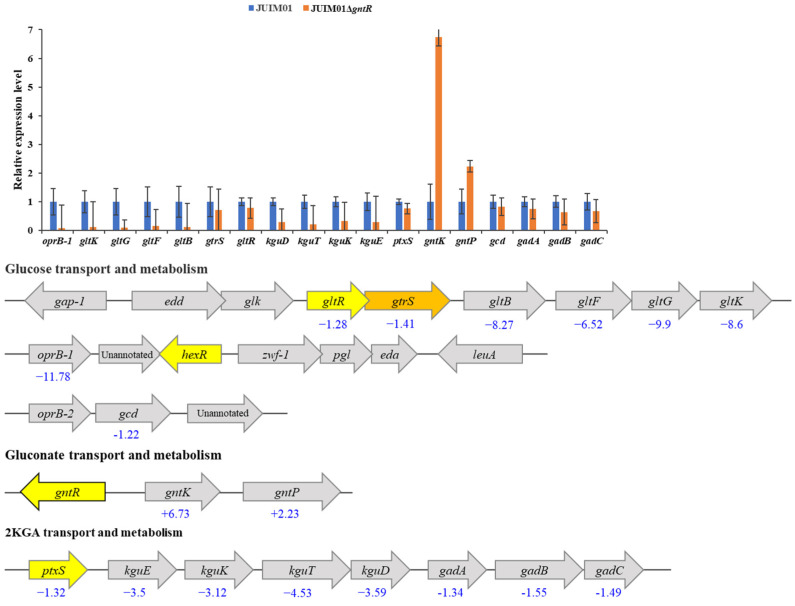
Comparison of the expression levels of related genes involved in glucose/gluconate/2KGA metabolism between *P. plecoglossicida* JUIM01 and JUIM01Δ*gntR*. The yellow and orange arrows represent the genes encoding transcriptional regulators.

**Figure 3 microorganisms-13-01395-f003:**
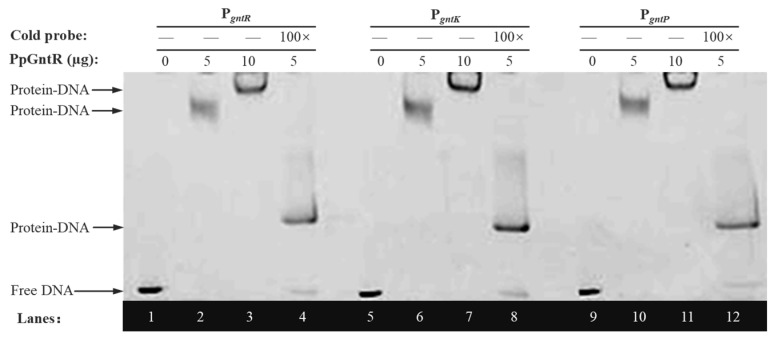
Electrophoretic mobility shift assays of P*_gntR_*, P*_gntK_*, and P*_gntP_* promoter fragments with PpGntR.

**Figure 4 microorganisms-13-01395-f004:**
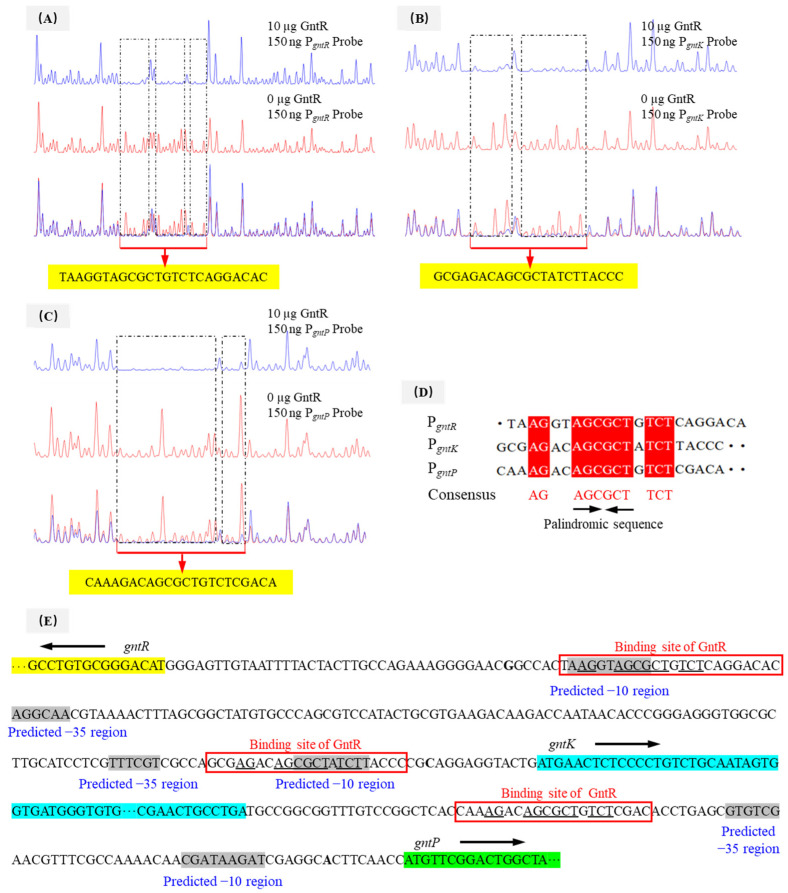
DNase I footprinting assays of (**A**) P*_gntR_*, (**B**) P*_gntK_*, and (**C**) P*_gntP_* promoter fragments with PpGntR, (**D**) analysis of conserved GntR-binding motif, and (**E**) prediction of the regulatory mechanism of GntR in *P. plecoglossicida*.

**Figure 5 microorganisms-13-01395-f005:**
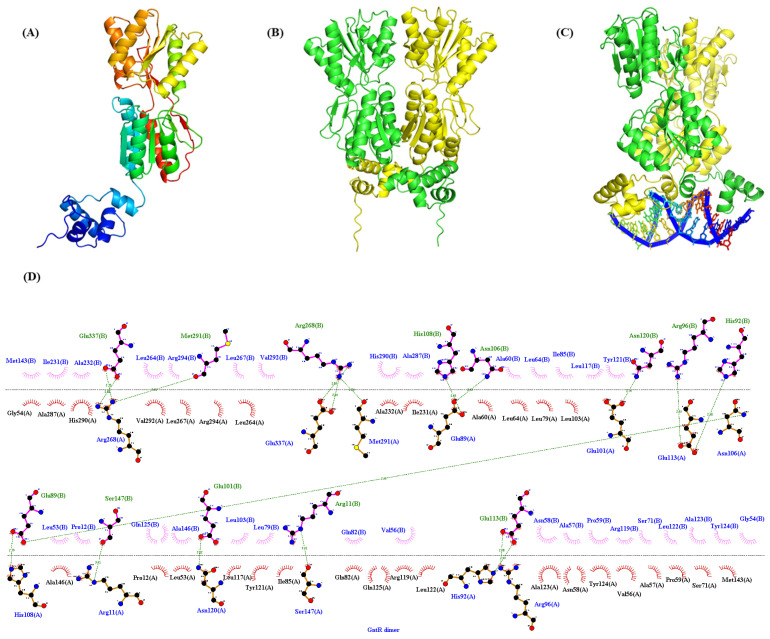
The tertiary-structure and protein-DNA binding prediction of GntR. (**A**) Monomer; (**B**) Dimer; (**C**) The binding of GntR dimer to the conserved binding motif (5′-AG-N_2_-AGCGCT-N-TCT-3′); (**D**) The dimerization interfaces of GntR.

**Table 1 microorganisms-13-01395-t001:** Strains and plasmids used in this study.

Strains and Plasmids	Description	Source
**Strains**		
*P*. *plecoglossicida* JUIM01	A 2-ketogluconate industrial producing strain	Our Lab
*E. coli* JM109	General cloning strain	TaKaRa
*E. coli* BL21(DE3)	General expressing strain	TaKaRa
*E. coli* JM109/pK18*mobsacB*-Δ*gntR*	JM109 containing vector pK18*mobsacB*-Δ*gntR*	This work
*P*. *plecoglossicida* JUIM01Δ*gntR*	*gntR*-knockout mutant of JUIM01	This work
*P*. *plecoglossicida* JUIM01Δ*gntR-gntR*	*gntR*-complemented strain of JUIM01Δ*gntR*	This work
*E. coli* JM109/pMD20-T-*gntR*	JM109 containing vector pMD20-T-*gntR*	This work
*E. coli* BL21(DE3)/pET-28a(+)	BL21(DE3) containing vector pET-28a(+)	Our Lab
*E. coli* BL21(DE3)/pET-28a(+)-*gntR*	BL21(DE3) containing recombinant vector pET-28a(+)-*gntR*	This work
**Plasmids**		
pK18*mobsacB*	Mobilizable *E*. *coli* vector, Kan^r^, Suc^s^	Our Lab
pET-28a(+)	Expression vector, carrying an N-terminal His-Tag/thrombin/T7-Tag and C-terminal His-Tag, Kan^r^	Our Lab
pMD20-T	T-vector, 2736 bp, Amp^r^, *lacZ*	TaKaRa
pBBR1MCS-2	*E. coli*-*Pseudomonas* shuttle vector, Kan^r^	Our Lab
pK18*mobsacB*-Δ*gntR*	pK18*mobsacB* containing incomplete *gntR* sequence of JUIM01	This work
pBB-*gntR*	pBBR1MCS-2 containing the *gntR* of JUIM01	This work
pMD20-T-*gntR*	pMD20-T containing the *gntR* of JUIM01	This work
pET-28a(+)-*gntR*	pET-28a(+) containing the *gntR* of JUIM01	This work

## Data Availability

The original contributions presented in the study are included in the article/Appendix A, and further inquiries can be directed to the corresponding authors.

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
