# Peer review of "Molecular Characterization of a Transcriptional Regulator GntR for Gluconate Metabolism in Industrial 2-Ketogluconate Producer Pseudomonas plecoglossicida JUIM01"

_microorganisms, 2025, doi:10.3390/microorganisms13061395_

Round 1

Reviewer 1 Report

Comments and Suggestions for Authors

The study "Molecular Characterization of the Transcriptional Regulator GntR in Gluconate Metabolism of the Industrial 2-Ketogluconic Acid Producer Pseudomonas plecoglossicida JUIM01" investigates the role of the transcriptional regulator GntR in gluconate metabolism by constructing a gntR deletion mutant and analyzing its effects on the transcription of gluconate metabolism-related genes. The authors further identified GntR binding sites on target promoters using DNase I footprinting assays.

The experimental design is well-structured, with appropriate controls, and the conclusions are well-supported by the presented data. The topic is relevant and significant, particularly for understanding metabolic regulation in industrial bacterial strains. However, the manuscript would benefit from improved clarity in English expression and addressing the following key points before publication:

Major Comments:

  1. Inconsistency in gntR identification (Lines 261–265):

The authors state that gntR was not found in P. plecoglossicida but later mention cloning a 1020-bp gntR sequence from P. plecoglossicida JUIM01. This discrepancy should be clarified.

  1. Inoculation method for E. coli (Lines 285–287):

The manuscript does not specify whether an overnight culture was used to inoculate LB medium. Clarifying the inoculation conditions would improve reproducibility.

  1. Figures 2, 3, and 4:

Adding arrows to indicate the GntR band would enhance readability, especially in gel images where multiple bands are present.

  1. Structural insights on GntR (Lines 318–331):

Given recent advances in protein structure prediction, using AlphaFold3 to generate a 3D model of GntR could provide valuable mechanistic insights (e.g., DNA-binding domain organization, potential dimerization interfaces). This addition would strengthen the discussion.

Minor Comment:

Line 376: Replace the second gntP with gntR for consistency.

Comments on the Quality of English Language

The English expression can be improved further.

Author Response

Comment 1: Inconsistency in gntR identification (Lines 261–265): The authors state that gntR was not found in P. plecoglossicida but later mention cloning a 1020-bp gntR sequence from P. plecoglossicida JUIM01. This discrepancy should be clarified.

Response 1: Thank you for your suggestion. The sentence “However, no searching results of gntR were observed in P. plecoglossicida, although some CDS sequences with a length of 669-1356 bp belonging to GntR family transcriptional regulator were searched” has been revised as “However, no gntR-like gene originating from P. plecoglossicida was present in this database”.

Comment 2: Inoculation method for E. coli (Lines 285–287): The manuscript does not specify whether an overnight culture was used to inoculate LB medium. Clarifying the inoculation conditions would improve reproducibility.

Response 2: Thank you for your suggestion. The inoculation conditions have been added in Materials and Methods, Section 2.4 and shown as “The recombinant E. coli BL21(DE3)/pET-28a(+)-gntR strains cultured overnight were inoculated into the LB broth containing 25 µg/mL of kanamycin at 1% inoculation, and were cultured at 37 °C and 200 rpm to an optical density at 600 nm (OD600 nm) of 0.5-1.0”.

Comment 3: Figures 2, 3, and 4: Adding arrows to indicate the GntR band would enhance readability, especially in gel images where multiple bands are present.

Response 3: Thank you for your suggestion. The arrows have been added in these figures to indicate the GntR bands.

Comment 4: Structural insights on GntR (Lines 318–331): Given recent advances in protein structure prediction, using AlphaFold3 to generate a 3D model of GntR could provide valuable mechanistic insights (e.g., DNA-binding domain organization, potential dimerization interfaces). This addition would strengthen the discussion.

Response 4: Thank you for your suggestion. The protein structure prediction using AlphaFold3 has been added in Section 2.2 and 3.5.

Comment 5: Line 376: Replace the second gntP with gntR for consistency.

Response 5: Yes, the sentence has been revised.

Comments on the Quality of English Language: The English expression can be improved further.

Response: Thank you for your suggestion. The English expression has been improved.

Reviewer 2 Report

Comments and Suggestions for Authors

The manuscript describes the characterization of a GntR regulator from Pseudomonas plecoglossicida JUIM01. The authors knocked out the gntR gene and observed changes in gene expression. They also cloned and expressed the gntR gene in Escherichia coli and carried out various biochemical characterizations, which allowed them to identify the promoters upstream of the gntR, gnuK, and gntP genes. The manuscript is clearly written, and the methodology is sound. However, I do have a few concerns regarding the manuscript that should be fixed.

  1. Lines 224-225: are the gntP and gntR swapped?
  2. Figure 2: is this the soluble protein (lysate supernatant) or the crude cell extract (lysate)? Please clarify in the figure legend.
  3. Figure 6- please include a description of the genes depicted as yellow arrows (regulators?) in the figure legend.
  4. Line 376- “The gntP gene showed an opposite..” Shouldn’t this be gntR? The opposite direction of the regulator is consistent with many gntR-like genes.
  5. The English is good, but could use some polishing from a native English speaker. There were also a few typographical errors (misspellings, punctuation, etc.).

Author Response

Comment 1: Lines 224-225: are the gntP and gntR swapped?

Response 1: Yes, the sentence has been revised.

Comment 2: Figure 2: is this the soluble protein (lysate supernatant) or the crude cell extract (lysate)? Please clarify in the figure legend.

Response 2: The figure shows the soluble proteins (lysate supernatants). The description has been supplemented in the figure legend.

Comment 3: Figure 6- please include a description of the genes depicted as yellow arrows (regulators?) in the figure legend.

Response 3: Yes, the yellow and orange arrows represent the genes encoding the transcription factors. The description has been supplemented in the figure legend.

Comment 4: Line 376- “The gntP gene showed an opposite..” Shouldn’t this be gntR? The opposite direction of the regulator is consistent with many gntR-like genes.

Response 4: Yes, the sentence has been revised.

Comment 5: The English is good, but could use some polishing from a native English speaker. There were also a few typographical errors (misspellings, punctuation, etc.).

Response 5: Thank you for your suggestion. The English expression has been improved.

Reviewer 3 Report

Comments and Suggestions for Authors

In their manuscript entitled “Molecular characterization of a transcriptional regulator GntR for gluconate metabolism in industrial 2-ketogluconic acid producer Pseudomonas plecoglossicida JUIM01” the authors characterized the regulation, the targets genes and the binding site of the transcriptional regulator of gluconate metabolism, GntR, in the industrial 2-ketogluconic acid producer Pseudomonas plecoglossicida JUIM01. Data obtained turned out to be quite similar to those obtained for GntR of Pseudomonas aeruginosa, so the study does not bring much novelty. It also bears many uncertainties and improper or missing controls. Furthermore english wording requires corrections in many places. My analysis and interpretation of the data is not in agreement with those of the authors. The authors should carefully check my proposals and think about them.

In the joined file my comments are in red in the text or in the margins and corrections of the text are in blue.

Comments on the Quality of English Language

English langage needs editing, some proposal were made.

Author Response

Comment 1: In their manuscript entitled “Molecular characterization of a transcriptional regulator GntR for gluconate metabolism in industrial 2-ketogluconic acid producer Pseudomonas plecoglossicida JUIM01” the authors characterized the regulation, the targets genes and the binding site of the transcriptional regulator of gluconate metabolism, GntR, in the industrial 2-ketogluconic acid producer Pseudomonas plecoglossicida JUIM01. Data obtained turned out to be quite similar to those obtained for GntR of Pseudomonas aeruginosa, so the study does not bring much novelty. It also bears many uncertainties and improper or missing controls. Furthermore english wording requires corrections in many places. My analysis and interpretation of the data is not in agreement with those of the authors. The authors should carefully check my proposals and think about them.

In the joined file my comments are in red in the text or in the margins and corrections of the text are in blue.

Response 1: Thank you for your suggestion. The manuscript has been improved according to your proposals shown in the joined file. The revisions have been made and responses are listed as follows:

(1) The abstract has been improved as your proposal.

(2) Section 3.1: the sentence “However, no searching results of gntR were observed in P. plecoglossicida, although some CDS sequences with a length of 669-1356 bp belonging to GntR family transcriptional regulator were searched” has been revised as “However, no gntR-like gene originating from P. plecoglossicida was present in this database”.

(3) Section 3.1: the hydropathy plot of the GntR was predicted using the Kyte & Doolittle method by ProtParam tool (https://web.expasy.org/protparam/). The result has been added in the supplementary materials (Figure S4), indicating the GntR is supposed to be hydrophobic with a grand average of hydropathicity (GRAVY) of 0.071.

(4) The gene gnuK has been changed to gntK throughout the manuscript.

(5) The words “gene(s)” after the italic names have been deleted as your suggestion.

(6) Section 3.2 and 3.3: We have shortened the contents and placed the figures and table in the supplementary materials as your suggestion.

(7) Section 3.4: the formulation has been improved as your proposal.

(8) Section 3.4: The 16S rRNA gene is a highly conserved component in the bacterial genomes, including Pseudomonas plecoglossicida, and its expression level generally remains relatively stable, making it suitable as a reference benchmark. In this study, we calculated the relative expression levels of 14 target genes normalized to this internal reference gene, thereby analyzing the relative changes in their expression before and after gntR knockout.

(9) Based on our experimental results (not shown in this study), the knockout of gntR did not significantly affect 2KGA production in Pseudomonas plecoglossicida. We therefore hypothesized that the conversion of glucose to gluconate (the precursor of 2KGA) was likewise not substantially limited.

(10) We indeed constructed a gntR-complemented strain. When cultured in medium containing glucose as substrate, no significant differences were observed in either cell growth or 2KGA production among the wild-type JUIM01, gntR knockout mutant, and gntR-complemented strain (data not shown in this study).

(11) The growth curves of the WT and GntR mutant strains grown in a glucose containing medium has been added (Figure S9) to determine whether the deletion of GntR affects growth.

(12) Descriptions of ptxS, gltR, and gtrS have been added in Section 3.4.

(13) Based on our analysis and speculation, the downregulated genes are likely not directly controlled by GntR. Taking the kgu operon as an example, its upstream region contains a 14-bp palindromic sequence (5′-TGAAACCGGTTTCA-3′) that can be recognized and directly regulated by the transcriptional regulator PtxS (Swanson et al., 2000; Sun et al., 2024). Therefore, we propose that the carbon metabolism in Pseudomonas is governed by a complex regulatory network, where multiple transcriptional regulators (such as GntR, PtxS, and GtrS-GltR) coordinately control substrate (glucose, gluconate, and 2KGA) transport, metabolic flux partitioning, and cellular growth homeostasis.

(14) Sorry, the levels of gcd and gad expression were not tested. We will study them in the subsequent research.

(15) Figure 2 (previous Figure 6): The colour codes represent the genes encoding the transcription factors. The description has been supplemented in the figure legend.

(16) Section 3.5: the formulation has been improved as your proposal.

(17) FAM stands for carboxyfluorescein, which has been added in Section 2.9 the first time it appears in the manuscript.

(18) Section 3.5: the results description of EMSA have been improved. The results (Figure 3) revealed specific interactions between PpGntR and its target promoters, with control lanes (Lane 1, 5 and 9) showing unshifted migration of FAM-labelled PgntR, PgntK and PgntP probes, confirming their integrity in the absence of PpGntR. Dose-dependent band shifts were observed upon addition of increasing PpGntR concentrations (5 μg in Lane 2/6/10; 10 μg in Lane 3/7/11), demonstrating protein-DNA complex formation. Binding specificity was further validated through competitive displacement assays (Lane 4/8/12), where 100-fold molar excess of unlabelled probe significantly reduced shifted complex intensity while restoring free probe migration.

(19) Section 3.5: sorry, we did not try to add only 10-fold cold probe. Only 100-fold molar excess of cold probe was used to make sure it is superfluous.

(20) As shown in Materials and Methods (Section 2.9), in the EMSA system, an excess of salmon sperm DNA (rich in random sequences) was added as nonspecific competitor DNA.

(21) Section 3.6 and Figure 4E (previous Figure 8E): yes, the -10 and -35 regions are only predicted results using bioinformatic tools. We carefully considered your suggestions. To avoid possible misunderstandings, we have removed the relevant analysis and discussion from the manuscript.

(22) Conclusion: the formulation has been improved as your proposal.

(23) The metabolic engineering strategy has been added in Section 3.4.

Comment 2: English langage needs editing, some proposal were made.

Response 2: Thank you for your proposal. The language has been improved.

Round 2

Reviewer 3 Report

Comments and Suggestions for Authors

See joined file

Comments on the Quality of English Language

Quality of english can still be improved

Author Response

Comment 1: 16S RNA is not a proper control for RT-PCR since it is not translated into protein and is a very stable RNA.

Response 1: Sorry, but we have a differing opinion from the reviewer. Here is our previous response: "The 16S rRNA gene is a highly conserved component in bacterial genomes, including Pseudomonas plecoglossicida, and its expression level generally remains relatively stable, making it suitable as a reference benchmark. In this study, we calculated the relative expression levels of 14 target genes normalized to this internal reference gene, thereby analyzing the relative changes in their expression before and after gntR knockout."

Relative quantitation is best applied when there are many genes to be tested across multiple samples, which relies on the assumption that the endogenous control gene does not vary under the experimental conditions (Ginzinger, 2002, doi:10.1016/S0301-472X(02)00806-8). Therefore, stability is an advantage, not a disadvantage, of using the 16S rRNA gene as a control. In fact, the 16S rRNA gene, as one of the most commonly used reference genes, has been widely used in qRT-PCR assays for long (Neretin et al., 2003, doi:10.1046/j.1462-2920.2003.00452.x; Gao et al., 2011, doi:10.1016/j.mimet.2011.03.008; Abishek N et al., 2023, doi:10.3390/cells12222596. Additional references can be provided if necessary).

Comment 2: The mutant should have been complemented.

Response 2: Yes, the description of the gntR-complemented strain has been added in the revised manuscript.

Comment 3: The authors should search the putative operator sequence upstream of the genes whose expression is down-regulated in the gntR deletion mutant or upstream of regulators known to regulate the expression of these genes.

Response 3: PtxS is a transcriptional regulator which specifically binds to a 14-bp palindrome (TGAAACCGGTTTCA) and negatively regulates the genes involved in 2KGA metabolism, including the gad operon encoding gluconate dehydrogenase (Gad), the kgu operon, and its own gene. In P. plecoglossicida JUIM01, this palindrome or similar sequence was identified upstream of the kgu operon and ptxS gene. However, no such sequence was found in the promoter region of the gad operon. gtrS together with gltR encode a two-component regulatory system GtrS-GltR, which regulates the target genes including gltB, and is required for glucose transport in P. aeruginosa. Daddaoua et al. reported the GltR operator site contains the consensus sequence tgGTTTTTc. However, the potential regulatory site of GltR identified by Xu et al. shows greater diversity. GtrS-GltR has not been reported in P. plecoglossicida, but sequences such as CTTTTTC, CTTTCTTGTTGTTATG, and CTTTTTG were found near the start codon of oprB-1, gltB, and oprB-2 (the first structural gene of the operon where gcd is located). Whether and how these genes are regulated by GtrS-GltR in P. plecoglossicida requires further investigation.

The analysis above has been added in Section 3.4.

Comment 4: The authors should test the level of expression of Gcd and Gad (line 45) enzymes responsible for the conversion of glucose to gluconate.

Response 4: The glucose dehydrogenase (Gcd) is encoded by gcd. The gluconate dehydrogenase (Gad) is encoded by the gad operon composed of three genes (gadA, gadB, and gadC). The relative expression levels of these four genes were modestly decreased (1.22-, 1.34-, 1.55-, and 1.49-fold, respectively) in JUIM01ΔgntR. The results have been added in Section 3.2 and Figure 2.

Comment 5: The authors should provide the necessary control when un-related cold DNA probe is added. The authors should comment on the faster migrating fragment observed when 100 fold cold specific probe is added?

Response 5: (1) As shown in Section 2.9, an excess of (2 µg) salmon sperm DNA was added in the EMSA system. Salmon sperm DNA, rich in random sequences, serves as an un-related cold DNA probe in the EMSA system.

(2) Our comment on the faster migrating fragment observed when 100-fold cold specific probe was added: The results (Figure 3) revealed specific interactions between PpGntR and its target promoters, with control lanes (Lane 1/5/9) showing unshifted migration of FAM-labelled PgntR, PgntK and PgntP probes in the absence of PpGntR. Dose-dependent band shifts were observed upon addition of increasing PpGntR concentrations (5 μg in Lane 2/6/10; 10 μg in Lane 3/7/11), demonstrating protein-DNA complex formation. Binding specificity was further validated through competitive displacement assays (Lane 4/8/12), where 100-fold molar excess of unlabeled probe competitively bound most PpGntR, thereby restoring the migration of the free labelled probes. Additionally, the system contained labeled probes that were unsaturatedly bound to PpGntR. The migration rate of these protein-DNA complexes is slower than that of the free probes but faster than that of the saturated protein-DNA complexes, causing them to appear in the middle position of the lanes.

Comments 6: Putative -10 and -35 promoter sequences should to be included in Figure 4 as previously requested.

Response 6: As the reviewer’s suggestion, putative -10 and -35 promoter sequences have been added. The -10 and -35 regions predicted by bioinformatics tools are shown in Figure 4, and another possible -10 region (the consensus sequence AGCGCT) proposed by the reviewer is mentioned in Section 3.6.

Comment 7: and some editorial amendments are also suggested below in red.

Response 7: Thank you for your suggestions. The language has been improved as your suggestions.

Comment 8: Therefore, a potential metabolic engineering strategy for P. plecoglossicida JUIM01 would involve the over-expression of GntR to repress the expression of the kgu operon (e.g., through gene knockout, RNA interference, or promoter engineering to render the kgu operon under GntR regulation). as well as that of the gene encoding the enzyme involved in the oxidation of gluconate into 2KGA. (Please give more details)

Response 8: Sorry, we have a different point of view, because the over-expression of GntR will not repress but contribute to the expression of the kgu operon. We don’t want to repress the oxidation of gluconate into 2KGA either. Our purpose is to improve 2KGA accumulation of the strain. Therefore, two strategies will be implemented: (1) over-expression of GntR to repress the expression of gntK and gntP, thereby reducing gluconate that enters the cytoplasm for catabolism; and (2) inhibition of the expression of the kgu operon to prevent 2KGA from entering the cytoplasm for catabolism through gene knockout, RNA interference, or promoter engineering to render the kgu operon under GntR regulation.